# Deficiency in Androgen Receptor Aggravates the Depressive-Like Behaviors in Chronic Mild Stress Model of Depression

**DOI:** 10.3390/cells8091021

**Published:** 2019-09-02

**Authors:** Yi-Yung Hung, Ya-Ling Huang, Chawnshang Chang, Hong-Yo Kang

**Affiliations:** 1Department of Psychiatry, Kaohsiung Chang Gung Memorial Hospital, and Chang Gung University College of Medicine, Kaohsiung 833, Taiwan; 2Graduate Institute of Clinical Medical Sciences, College of Medicine, Chang Gung University, Kaohsiung 833, Taiwan; 3George Whipple Lab for Cancer Research, Departments of Pathology, Urology and Radiation Oncology, and The Wilmot Cancer Center, University of Rochester Medical Center, Rochester, NY 14646, USA; 4Department of Obstetrics and Gynecology, Kaohsiung Chang Gung Memorial Hospital, Kaohsiung 833, Taiwan

**Keywords:** major depressive disorder, chronic mild stress, miR-204-5p, BDNF, androgen receptor

## Abstract

While androgen receptor (AR) and stress may influence the development of the major depressive disorder (MDD), the detailed relationship, however, remains unclear. Here we found loss of AR accelerated development of depressive-like behaviors in mice under chronic mild stress (CMS). Mechanism dissection indicated that AR might function via altering the expression of miR-204-5p to modulate the brain-derived neurotrophic factor (BDNF) expression to influence the depressive-like behaviors in the mice under the CMS. Adding the antiandrogen flutamide with the stress hormone corticosterone can additively decrease BDNF mRNA in mouse hippocampus mHippoE-14 cells, which can then be reversed via down-regulating the miR-204-5p expression. Importantly, targeting this newly identified AR-mediated miR-204-5p/BDNF/AKT/MAPK signaling with small molecules including 7,8-DHF and fluoxetine, all led to alter the depressive-like behavior in AR knockout mice under CMS exposure. Together, results from these preclinical studies conclude that decreased AR may accelerate the stress-induced MDD via altering miR-204-5p/BDNF/AKT/MAPK signaling, and targeting this newly identified signaling may help in the development of better therapeutic approaches to reduce the development of MDD.

## 1. Introduction

Major depressive disorder (MDD) is highly associated with marked personal, social and economic impact, and the susceptibility of MDD is affected by the interactions of multiple functional genetic variants and environmental factors. MDD has the greatest impact of all diseases on disability in the U.S, and it is the third leading cause of disability in Europe [1]. While the prevalence of MDD is higher in adult females than in adult males [2], more older men than older women reported depressive symptoms and the largest proportion of MDD patients was found among men in the age group 75–80 years [3]. Under low plasma testosterone levels, older men often exhibit depressive symptoms [4,5], and results from androgen replacement therapy indicated that older men receiving testosterone showed better moods and lower severity of depressive symptoms than those who received placebo [6]. These clinical evidences strongly suggest that androgens may have protective effects on adult men against MDD.

The biological actions of androgens are mediated by a ligand-dependent nuclear transcription factor, the androgen receptor (AR). An inverse relationship between AR transcriptional activity and the number of polyglutamine repeats in the AR transcriptional activation domain is well documented [7,8]. A previous study examining the relationship between the length of polyglutamine repeats and MDD indicated that the length of CAG trinucleotide repeats (CAG repeats) in the AR gene could predict the severity of MDD in young men [9]. Other studies also showed that prostate cancer patients receiving complete androgen blockade with the combination of flutamide and leuprolide might have significantly increased depression [10,11]. Recent studies reported that depression was observed in 36% of patients with complete androgen insensitivity syndrome due to loss of AR activity [12,13]. The amount of AR mRNA in the paraventricular nucleus of the human brains was decreased by 2.7-fold in the depressed patients as compared to the controls in a postmortem study [14]. Furthermore, AR/ERα expression index in the hippocampus were reported to regulate depressive-like behavior in rats [15]. Together, these studies suggest that loss of AR activity may be associated with increasing the risk of MDD.

Accumulating evidence suggests that brain-derived neurotrophic factor (BDNF) is associated with the pathophysiology of MDD. The plasma/serum levels of BDNF were decreased in MDD patients compared with healthy controls [16,17] and the decreased BDNF level was also detected in the hippocampus from suicide victims and mice under chronic mild stress [18,19,20]. In agreement with the associations between the BDNF Val66Met SNP and depression in men, but not in women [21,22], mice with conditional BDNF knockout in the forebrain also displayed gender differences in depression-related behaviors [23]. Androgen insensitive male, but not female rats, were found to display increased anxiety-like behaviors [24]. In addition, testosterone treatment has been shown to be effective on spine maturation in the hippocampus CA1 area of gonadectomized male mice via regulation of BDNF [25]. However, the molecular mechanisms of androgen/AR-mediated BDNF with its receptor tropomyosin receptor kinase B (TrkB) signaling in male mice under stress remain to be determined.

The microRNAs (miRNA, miR) are non-coding RNA molecules with 20–22 nucleotides, which modulate gene expression at the post-transcriptional level and brain specific miRNAs were found to play important roles in neurotrophin expression [26]. BDNF was predicted to be a miR-204 target from several miRNAs target prediction algorithms, and miR-204 was proven experimentally to bind to the predicted site in the 3′UTR of the BDNF gene [27]. Recently, it was reported that loss of miR-204 resulted in BDNF/TrkB overexpression and activation of the AKT/mTOR/Rac1 signaling pathway in cancer cells [27]. However, the interaction between BDNF and brain-specific miRNAs in MDD and other brain diseases is an area of recent interest [28].

Here we found that decreased AR may accelerate the stress-induced MDD via altering the miRNA-204-5p/BDNF/AKT/MAPK signaling, and targeting this newly identified signaling with small molecules may help in the development of better therapeutic approaches to reduce the development of MDD.

## 2. Materials and Methods

### 2.1. Animals 

All animal studies were handled in accordance with the “Guide for the Care and Use of Laboratory Animals” (National Institutes of Health publication). All procedures for testing and handling were approved by the Committee on Animal Resources of Kaohsiung Chang Gung Memorial Hospital. The strains of the mosaic founder mice were C57BL/6 and 129Sv background and Floxed AR/AR female mice were established and the procedures used for genotype determination as described in our previous publication [29,30]. The ACTB-Cre+/+ mice were purchased from Jackson Lab. To establish wild type (WT) and AR-knocked out (ARKO) mice, the floxed AR/AR female mice were bred with ACTB-Cre+/+ male mice to obtain the floxed AR+/y ACTB-Cre+/− as ARKO mice and the AR +/y ACTB-Cre +/− as WT control. All mice were group-housed in our colony at Kaohsiung Chang Gung Memorial Hospital with a 12/12 light/dark cycle, lights on at 0700. For the behavioral experiments, adult male mice (age = 10 weeks) were used in all tests and comparisons were made to age-matched littermate control cohorts. All tests were carried out in a behavioral suite adjacent to the holding room in the mouse facility at Kaohsiung Chang Gung Memorial Hospital. For behavioral tests requiring video-tracking, a video was acquired at 4 fps with 240 × 240-pixel spatial resolution using ethovision XT (Noldus, Wageningen, The Netherlands).

### 2.2. Behavior Experiment 

#### 2.2.1. Chronic Mild Stress (CMS) Protocol 

In the CMS paradigm, the animals were exposed to several stressors for 6 weeks (Figure 1A) and 2 weeks (Figure 1C). (The stressor schedule followed in this study (S1) was adapted from a previous protocol for mice [31] and rats [32]. The males in the unstressed groups were under handling and storage conditions comparable to the stressed animals.

#### 2.2.2. Sucrose Preference Test 

The protocol of sucrose preference test with 1% sucrose solution and forced swimming test was adapted from a previous study [31]. Mice were individually housed with two weighed water bottles (bottle “A” and bottle “B”). Water bottles were fitted with bottle stoppers containing one-balled sipper tubes. Bottle “A” contained 1% sucrose, and bottle “B” contained drinking water. The consumptions of sucrose and drinking water were measured after an 8 h test. Then the bottles A and B were switched to avoid a side bias and the consumptions were checked again. Sucrose preference was calculated as a percentage of the volume of sucrose intake over the total volume of fluid intake and averaged over the 4 days of testing. 

#### 2.2.3. Forced Swimming Test

Glass beakers (13 cm diameter × 24 cm high) were filled with fresh water (23–25 °C) to a depth of 12 cm. Mice were placed into the test beaker and were unable to escape or rest by touching the bottom of the beaker. Sessions were video recorded for 5 min and analyzed offline with ethovision XT (Noldus Information Technology, Wageningen, the Netherlands). Mice were scored by visual inspection for immobility defined as motionless floating in the water.

#### 2.2.4. Fluoxetine and 7,8-Dihydroxyflavone (7,8-DHF) Treatments for ARKO Mice

Fluoxetine (Sigma, Taufkirchen, Germany) was dissolved in sterile phosphate-buffered saline (PBS) and administrated i.p. daily at 10 mg/kg of body weight [33] for 14 days. The 7,8-DHF (Abcam, Burlingame, CA, USA) was dissolved in dimethyl sulfoxide (DMSO) and administrated i.p. twice per day at 5 mg/kg or 20 mg/kg of body weight for 14 days (Figure 4A).

### 2.3. Immunohistochemistry (IHC)

BDNF, pAKT and p-p38 staining were performed on 5 μm paraffin brain sections (sagittal, lateral 1.10–1.95mm). The sections were deparaffinized in xylene and then hydrated in 100–70% alcohol. Sections were incubated with the anti-BDNF (5 μg/mL, AB1534SP, Millipore), anti-pAKT(10 μg/mL Bioss #bs-5209R), or p38 MAPK (Thr180 + Tyr182) (10 μg/mL Bioss #bs-2210R) antibodies for 24 h at 4 °C. The sections were then washed three times with 0.2% Triton X-100 in PBS and then incubated with Dako REAL EnVision/HRP, Rabbit/Mouse (ENV) detection kit for 30 min at room temperature, followed by a color reaction using Dako REAL DAB+ chromogen for 1 min for BDNF, pAKT, and p-p38.

### 2.4. In Situ Hybridization (ISH) for BDNF 

In situ probe generation: The probe sequence was as follows: CAG TTG GCC TTT GGA TAC CGG GAC TTT CTC TAG GAC TGT GAC CGT CCC.

Coronal sections (5 μm) were mounted on poly-l-lysine-coated slides and allowed to air-dry for 16 h at 45 °C and were then de-paraffinized and rehydrated. The BioTnA Biospot Kit protocol was applied to the sections. The ISH was completed after applying DAB Solution to the slides for 1 min and the slides washed for 2 min under running tap water. We then counted the cells with positive staining and calculated the density.

### 2.5. Laser Capture Microdissection, and RNA Isolation 

LCM was performed with Veritas Automated Laser Capture Microdissection (LCM) System (Thermo Fisher Scientific, Inc., Waltham, MA, USA). A total of 10–15 series of sections were dissected for each sample. The RNA of microdissected cells was extracted using the PicoPure RNA isolation kit (Life Technologies, Carlsbad, CA, USA). The isolated RNAs were reversely transcribed using QuantiTect Reverse Transcription Kit (Qiagen, Hilden, Germany) following the manufacturer’s instructions. The cDNA was then pre-amplified using the TaqManPreAmp Master Mix (Life Technologies) according to the manufacturer’s instructions.

### 2.6. Cell Culture 

The embryonic mouse hippocampus, mHippoE-14, cell line was obtained from Cedarlane and maintained at 2 × 10^5^ cells/mL in RPMI 1640 media supplemented with 2 mM l-glutamine, 1.5 g/L sodium bicarbonate, 4.5 g/L glucose, 10 mM HEPES, 1.0 mM sodium pyruvate, and 10% fetal bovine serum (FBS). All media contained 1.5 μg/mL penicillin/streptomycin/neomycin, and cells were incubated at 37 °C in a humidified 5% CO_2_ atmosphere. In corticosterone and flutamide experiments, mHippoE-14 cells were seeded in 6-well plate treated with 1 μM flutamide for 48 h, and then treated with 100 ng/mL corticosterone. After 48 h, cells were harvested for further analysis. RNAs from harvested cells were extracted using Quick-RNA MiniPrep (Zymo Research, Orange, CA, USA).

### 2.7. Transient Transfection with siRNA 

ON-TARGET plus SMART pool small interfering RNA (siRNA) against mouse miR-204-5p and ON-TARGET plus non-targeting siRNA pool were purchased from Dharmacon. The TransIT-X2 Dynamic Delivery System (Mirus Corp., Madison, WI, USA) was used for siRNA transfection, as described by the manufacturer. In corticosterone treatment and miR-204-5p knockdown experiments, the mHippoE-14 cells in 6-well plates were transfected with 30 nM miR-204-5p siRNA or control siRNA. After 2 h transfection, cells were treated with 100 ng/mL corticosterone (Sigma Chemical Co., St. Lous, Mo, USA) or ethanol for 72 h. In flutamide treatment and miR-204-5p knockdown experiments, the mHippoE-14 cells in 6-well plates were treated with 1 μM flutamide for 48 h and then were transfected with 30 nM miR-204-5p siRNA or control siRNA for 24 h.

### 2.8. Quantitative Real-Time PCR (qRT-PCR) 

Mouse RNAs were reversely transcribed using QuantiTect Reverse Transcription Kit (Qiagen) following the manufacturer’s instructions. cDNA for microRNA were synthesized using the TaqMan Micro RNA Transcription Kit (Life Technologies) according to the manufacturer’s instructions. The qRT-PCR was performed using the following sets of primers: Human AR 5′-TCA CCG CAC CTG ATG TGT G-3′ (sense) and 5′-ACA TGG TCC CTG GCA GTC TC-3′ (antisense); human GAPDH, 5′-TGC ACC ACC AAC TGC TTA GC-3′ (sense) and 5′-GGC ATG GAC TGT GGT CAT GAG-3′ (antisense); mouse BDNF, 5′-TGA AGG GGC ATA GAC AAA AGG-3′ (sense) and 5′-CTT ATG AAT CGC CAG CCA ATT CTC-3′ (antisense); and mouse GAPDH, 5′-GCA CAG TCA AGG CCG AGA AT-3′ (sense) and 5′-GCC TTC TCC ATG GTG GTG AA-3′ (antisense), as well as miR-204-5p (Life Technologies, #000508), and U6 (Life Technologies, #001973). Relative expression levels of target genes in each sample were calculated based on the threshold cycle (CT), where the difference in CT (−ΔCT) used to represent relative expression of clinical samples was defined as CTGAPDH – CT sample. The 2−ΔΔCT method was used to calculate relative changes in expression of target genes for cell assays, where ΔΔCT = ΔCT treatment group − ΔCT control group.

### 2.9. Statistical and Data Analysis 

All sections used for IHC staining and ISH were microphotographed together in order to diminish the difference as much as possible. Data related to the IHC and ISH were analyzed with optical intensity staining using Image-Pro Plus software. The cell density of BDNF protein immunoreactive cells and BDNF RNA positive cells were counted by using 3-point scale [34]. The cell density was defined as positive cell/cells distribution region. The immunoreactive score gives a range as a product of multiplication between positive cells proportion score and staining intensity score (0–3). An independent T-test was used to analyze the differences in sucrose preference, forced swimming test, IHC intensity in animal models, and mRNA expression in the cell lines. For all comparisons, data are presented as mean ± SEM and *p* < 0.05 or *p* < 0.01 was used as a criterion for statistical significance.

## 3. Results

### 3.1. Loss of AR Accelerated the CMS-Mediated Depressive-Like Behavior in Mice 

To link the function of AR to the CMS-mediated depressive-like behavior in male mice, we first set up the protocol of CMS model of depression in wild type (WT) mice under CMS in different periods of time, and compared to the control group of non-stressed WT mice by Sucrose preference test (Figure 1A). It is well documented that CMS-exposed-mice with C57BL6 background require 3-weeks or longer CMS procedure to develop depressive–like behavior [35]. In agreement with previous finding, we found that sucrose preferences were decreased by 20% in CMS-exposed-mice after 5-weeks CMS procedure compared with control mice (*p* < 0.05; Figure 1B). Notably, the AR-knocked out (ARKO) mice had an early onset of depressive–like behavior with significantly decreased Sucrose preference compared to littermate WT mice after 2-weeks CMS procedure (Figure 1C,D). In the forced swimming test, the immobilization duration was significantly longer in the ARKO mice with CMS than in the ARKO without CMS exposure (Figure 1E). Together, results from Figure 1A–E suggest that loss of AR accelerated the CMS-mediated depressive-like behavior in ARKO mice.

### 3.2. Mechanism Dissection of Why Loss of AR Accelerated the CMS-Mediated Depressive-Like Behavior: Via Decreasing the BDNF Expression in the CA1 Region of Mouse Hippocampus in Response to CMS

The expression of AR has been reported to localize mainly in pyramidal neurons from hippocampal CA1 area and scarcely in some neurons from CA3 area and dentate gyrus [36]. Moreover, BDNF has also been demonstrated to be significantly decreased in the hippocampus CA1 area of CMS-treated mice [37]. To further dissect whether the loss of AR affect the BDNF expression in CMS-treated mice, we examined the BDNF protein expression in the CA1 and CA3 area of hippocampus from WT and ARKO mice under CMS procedures for 2 weeks by using IHC staining (Figure 2A–L). The results revealed that the density of BDNF immunoreactive cells was significantly decreased in the CA1 area (Figure 2E–H,U) but not in CA3 area (Figure 2I–L,V) of the hippocampus of CMS-treated ARKO mice.

We also applied the ISH assay to examine the BDNF mRNA expression and localization in the hippocampus and found that the cell density of BDNF mRNA positive cells in the CA1 area was significantly decreased after the CMS procedure (Figure 2S,T,W). However, there was no difference between in WT mice with or without CMS exposure (Figure 2Q–T,W). 

Together, results from Figure 2 suggest that loss of AR may function via decreasing the BDNF expression in the CA1 region of mouse hippocampus in response to CMS to accelerate the CMS-mediated depressive-like behavior in the mice.

### 3.3. Mechanism Dissection of How Androgens/AR Increases the BDNF Expression: Via Altering the miRNA-204-5p Expression 

Recent studies demonstrated BDNF signaling, via its TrkB receptor. It cooperatively interacts with androgens to maintain somal and dendritic morphology of bulbocavernosus (SNB) motoneurons [38] and the androgen-dependent loss of muscle BDNF mRNA in two mouse models of spinal bulbar muscular atrophy [39]. In addition, however, the mechanisms of how androgen/AR signaling regulates BDNF in the hippocampus regions remain unclear. To further dissect the molecular mechanisms of how AR can increase the BDNF expression, we treated mHippoE-14 cells, with stress-hormone corticosterone and antiandrogen flutamide. The results showed that BDNF mRNA expression was decreased under treatment of corticosterone and flutamide in mouse hippocampus cell line mHippoE-14 (Figure 3A) as well as neuroblastoma cell line Neuro-2A cells (data not shown). 

Recent studies indicated that decreasing the miR-204 expressing resulted in increased the BDNF/TRKB expression [27], and the miR-204 expression was found to be increased under stress exposure [40] or down-regulated by androgens/AR [41]. Thus, we were interested to see if AR may regulate the BDNF via altering the miR-204 in mHippoE-14 cells. We found that miR-204-5p mRNA expressions were increased after treating with the corticosterone and flutamide (Figure 3B). 

Importantly, we also applied the interruption approach using siRNA of mir-204 (si-mir-204 5p) to examine its impact on androgens/AR effects on BDNF expression in the mHippoE-14 cells. The results revealed that corticosterone- (Figure 3C) and flutamide- (Figure 3D) decreased BDNF expression was reversed after knocking down miR-204-5p. Compared to wild-type naïve mice, there is a trend of increasing miR-204-5p in wild-type mice after receiving 2 weeks of CMS, miR-204-5p is significantly increased in naïve and stressed ARKO mice (Figure 3E).

Together, results from Figure 3A–E suggest that androgens/AR may increase the BDNF expression via altering the miR-204-5p expression.

### 3.4. Targeting the AR/BDNF/TrkB-Mediated pAKT/p-p38 Signals with 7,8-DHF and Fluoxetine Attenuates the Depressive-Like Behavior in ARKO Mice with CMS Exposure 

BDNF/TrkB signaling has been shown to activate AKT and MAPK kinase cascades [42]. To target this newly identified AR/BDNF/TrkB-mediated pAKT/p-p38 signals in the ARKO mice with CMS, we treated the mice with the low-dose (5 mg/kg) and high-dose (20 mg/kg) of 7,8-DHF, a TrkB receptor agonist, along with fluoxetine, an anti-depressant that has been shown to increase BDNF [43], (Figure 4A). The results revealed that high-dose of 7,8-DHF and fluoxetine were able to attenuate the depressive-like behavior in ARKO mice with CMS in the Sucrose preference test (Figure 4B). Results from the IHC assay also revealed that much higher expression of pAKT and p-p38 were found in ARKO mice treated with high-dose (20 mg/kg) of 7,8-DHF and fluoxetine (Figure 4C–E).

Together, these results from Figure 4A–E suggest that targeting the AR/BDNF/TrkB-mediated pAKT/p-p38 signals with 7,8-DHF and fluoxetine attenuates the depressive-like behavior in ARKO mice with CMS exposure.

## 4. Discussion

Previous studies indicated the positive linkage between androgen and depression symptom severity [4,44], and other studies also reported that AR gene polymorphism was associated with vulnerability to major depression [9,45] or anxiety [46]. However, the molecular mechanisms of androgen/AR actions on the development and progression of depression remain unclear. According to epidemiological research, genetic predispositions play an important role in development of depression. Resilience and resistance to stress are common; therefore, although a certain extent of stress might exist during puberty, having a certain extent of stress without hormone imbalance does not imply that a mental disorder will develop. However, having a hormone imbalance or genetic predispositions of hormone receptor signaling does weaken resistance to stresses received during aging, and without the ability to recover, a mental disorder is triggered. Involvement of stress and AR signaling in the onset of depression was investigated in this current study. Here we propose a two-hit hypothesis or multiple-hit hypothesis in depression development similar to the Knudson hypothesis of cancer carcinogenesis (Figure 5). AR-deficient mice do not have reduced activity or response to the forced swim test compared to the WT mice and these mice also do not exhibit depressive-like behaviors. However, when mice with the disrupted AR gene received isolated chronic mild stress, depressive-like abnormal behaviors, the increased expression of microRNA-204-5p, and decreased gene expression of BDNF in the hippocampus were observed. 

The effects of androgen/AR signals on behavior and psychopathology have been proposed to be of value in preventing anhedonia in middle-aged males [44,47]. Male patients with congenital hypogonadotropic hypogonadism had overall significantly higher rates of developing depression and anxiety [48]. In androgen insensitive rats, an increased anxiety-like behavior [49] has negative impacts on the physical symptoms of MDD [50]. While testosterone levels and AR gene polymorphism have been shown to predict specific symptoms of depression in young males [9], the association of AR mRNA expression in adult male patients with MDD remain unclear. To confirm the potential linkage of AR expression to the MDD in men, we also compared the AR mRNA expression in peripheral blood mononuclear cells (PBMCs) from male MDD patients with healthy control subjects. The results revealed that AR mRNA expression is down-regulated in male patients with MDD comparing with health controls (Appendix A). This finding is consistent with a previous post-mortem study that reported the amount of AR mRNA in the paraventricular nucleus of the hypothalamus was decreased by ~2.7-fold in the depressed patients as compared to the controls [14]. 

Previous studies have demonstrated the consistent relationship between loss of androgen and depression symptom severity [4,44]. Several studies also reported that AR gene polymorphism was associated with vulnerability to major depression [9,45] or anxiety [46]. However, the molecular mechanisms of androgen/AR actions on the development and progression of depression remain unclear. Here we showed that loss of AR in male mice resulted in the development of early onset depressive-like behaviors and the decrease in BDNF expression within 2 weeks, whereas in the control C57BL6 mice this required 4–7 weeks to develop in the presence of stress [51,52]. To our knowledge, this is the first study to explicitly demonstrate that the loss of AR function exacerbates the depressive-like behavior in mice, as observed following CMS. 

Immunoreactivity of AR has been reported to localize mainly in hippocampal CA1 pyramidal neurons and scarcely in CA3 and dentate gyrus neurons [36], brain regions that show shrinkage in patients with MDD [53] and chronic stress can suppress neuronal function, especially synaptic plasticity in CA1 and CA3 areas of the hippocampus in rats [54,55]. Importantly, androgen has been suggested to modulate neuronal synaptic plasticity [56,57] and maintain the structure of the hippocampus, specifically in the CA1 area of the hippocampus [58]. This concept was further supported by the evidences showing that gonadectomy in mice induced a significant decrease in the protein levels of BDNF and PSD-95 in the CA1 area, which were prevented by androgen replacement therapy [25,58]. In agreement with the previous findings that the BDNF protein level in the CA1 of the hippocampus is decreased under CMS [59] and the relative gene expression of hippocampal BDNF in mice is positively correlated with specific neuron’s populations in hippocampus, such as GABA-containing neurons [60] which are involved in depressive level regulation [61], here we further demonstrated that the loss of AR in mice was not only evident as the change in the total level of the protein, but also as changes in mRNA expression in the CA1 of the hippocampus under CMS. There are some limitations in our current study. First, DAB used for immunohistochemistry and in situ hybridization does not follow the Beer-Lambert law that describes the linear relationship between the concentration of a compound and optical density [62]. Second, the age of mice in this study does not match human data. However, major depressive disorder in young men [9] and male adolescents [63] were reported to be important and associated with AR, our study is focused on age-dependent pathophysiological mechanisms. The aging process can lead to the impairment of brain functions and a decrease in whole brain volume in elderly when compared with young adults, especially in brain regions related to cognition has been reported [64], it would be better that our experiments could be reproducibly conducted chronic mild stress protocol in older animals with relative low level of testosterone and BDNF expression. Third, we cannot exclude the possibility that AR-mediated BDNF actions is not specifically in the CA1 area of the hippocampus, due to the limitation of the global ARKO mice used in this study. Given the importance of androgens/AR signaling axis in the course of brain development, there are potential consequences due to a global lack of AR but not tissue-specific AR deficiency in the CA1 area that leads to depressive–like behaviors in mice. While tamoxifen-dependent Cre recombinase, also known as Cre-ER recombinase, tamoxifen (TAM) has been used as a popular tool to activate the Cre to generate time- and tissue-specific mouse mutants, TAM is also an active selective estrogen receptor modulator (SERM) that has been shown to have anti-depressive activity in mice under chronic stress [65]. In order to minimize the effects of TAM on modulating the estrogen actions on brains, we are on the way to optimize the TAM dose for Cre recombinase activation and generate hippocampal CA1 specific AR knockout mice by crossing floxAR mice with Cre transgenic mice, T29–1, which have the capacity to mediate Cre/loxP recombination exclusively in the CA1 pyramidal cells [66].

While impacts of androgens on promoting neonatal neurogenesis in the developing rat hippocampus had been studied [67], studies of whether the loss of AR affects the development of hippocampus remain limited. The volume of the CA1 stratum radiatum and CA1 spine synapse density of adult male Tfm rat containing mutant AR are indistinguishable from wild type male rats [68]. Our data also supported that there was no difference between the WT and ARKO mice without CMS procedure, since the cell density of BDNF-positive cells in the hippocampus were maintained similar levels in ARKO mice compared to WT control mice (Figure 2U). In addition, ARKO mice did not develop depressive-like behavior and had maintained brain BDNF level at 10 weeks of age without CMS (data not shown). One of the possibilities is that the loss of AR may not be sufficient to induce depressive-like behaviors and the reduction in BDNF level may be compensated for by other adaptive stress response signaling pathways, such as CREB and cAMP-regulated transcriptional co-activators (CRTCs) signaling [69], which act as sensors for hormonal and metabolic signals and protect against AR deficiency-mediated neuron degeneration in adult ARKO mice. Although the expression and activity of CREB has not yet been explored in ARKO mice, the interactions between AR and CREB signaling on coordinating BDNF gene expression in neuronal cells under stress are of interest to be further studied.

Castration of adult male rats caused the dendrites of androgen-sensitive motoneurons of the spinal nucleus of the bulbocavernosus (SNB) to retract and the frequency of motoneurons intensely immunolabeled for TrkB receptors was regulated by the presence of testosterone [70]. Here we also demonstrated that 7,8-DHF, a TrkB receptor agonist that mimics the effect of BDNF, prevented the depressive-like behavior in ARKO mice subjected to CMS. These data indicate that BDNF/TrkB may be a critical downstream target of the androgen/AR actions in response to stress.

Previous studies have shown that the overexpression of BDNF/TrkB might not only provide an autocrine survival pathway for neurons, but also has been implicated in poor prognosis in several tumors [71,72]. Interestingly, miR-204 has an equally important role in tumorigenesis, and has been shown to be repressed by AR signaling in prostate cancer cells [41], but was induced by stress in animal studies [40]. In addition, the chromosomal locus containing miR-204 is frequently lost in cancers, resulting in its lower expression [73] and resulted in BDNF/TrkB overexpression and activation of the AKT/mTOR/Rac1 signaling pathway [27]. Several studies have shown that an increase in miR-204-5p in hippocampus neuronal cells is associated with senescence during aging [74]. Androgen deficiency or deprivation has also been shown to induce senescence in cancer cells, but other neuronal types as well [75]. It is possible that the loss of AR activity both in vivo and in vitro and the subsequent upregulation of miR-204-5p may be also related to neuronal cells senescence in hippocampus. Together, these data suggest that AR-mediated miR-204 may play a causal role in suppressing miR-204-5p/BDNF and play important roles for MDD disorders. Here we found that the increasing expression of miR-204-5p was observed in ARKO mice under CMS and this is the first evidence to show that the accumulation effect of miR-204 from combining stress with the loss of AR activity is significantly associated with depressive-like behavior. Furthermore, we found that corticosteroid and flutamide treatment suppressed the BDNF gene expression and increased the miR-204 expression in mHippoE-14 mouse hippocampus cells (Figure 3A,B). Knockdown of miR-204 expression reversed the effect of stress hormone and AR antagonist–mediated BDNF suppression (Figure 3C,D). Interesting, we noticed while there is an increasing miR-204-5p expression observed in naïve and stressed ARKO mice (Figure 3E), the expression of BDNF is decreased dramatically only in ARKO mice after CMS but not in controls and naïve ARKO mice (Figure 2U,W). Several factors have been shown to affect BDNF level under chronic mild stress such as impairment of glutamate/GABA presynaptic release, BDNF mRNA trafficking [76], and reduced acetylation levels of histone H3 at the promoter of exons I, IV, and VI of BDNF [77]. MiR-204-5p is one of the mediators that could down-regulate the BDNF expression and may not account for all the effects on regulating the BDNF level. According to previous findings, CMS-exposed-mice require 3-weeks or longer CMS procedure to develop depressive–like behavior [35,78,79], and 8-weeks procedure to decrease level of BDNF [80]. As shown Figure 1C,D, the ARKO mice had an early onset of depressive–like behavior with decreasing level of BDNF compared to littermate WT mice after 2-weeks CMS procedure. It is possible that 2-weeks CMS procedure is not enough to decrease the BDNF level in WT mice and the increasing level of miR-204-5p is also not enough to decrease the BDNF level in naïve ARKO mice. Since the increased miR-204-5p expression was found in mice under stress exposure [40], the combination of both AR deficiency and chronic stress may be required for boosting the effects of miR-204-5p to dramatically decrease the BDNF level in ARKO mice only after 2-weeks CMS procedure. Together, these results unify the evidences regarding the anti-depression effects of AR and suggest that AR may have protective roles in antagonizing stress-induced depressive-like behaviors through down-regulation of miR-204 to promote BDNF expression. 

## 5. Conclusions

In conclusion, the present study describes a depression model in response to stress-based gene-hormone/environment interactions. Our work identifying the miR-204 that is directly affected by AR during CMS has revealed novel protective mechanisms of AR actions on depression. Loss of AR accentuates the noxious effects of CMS on the development of depressive-like phenotypes and 7,8-DHF, a TrkB receptor agonist, could ameliorate the depressive-like behavior. Future development of novel antidepressant agents targeting AR, which, together with other testosterone interventions, could allow the AR/BDNF/TrkB axis to function properly for optimizing the management of depressive-related diseases.

## Figures and Tables

**Figure 1 cells-08-01021-f001:**
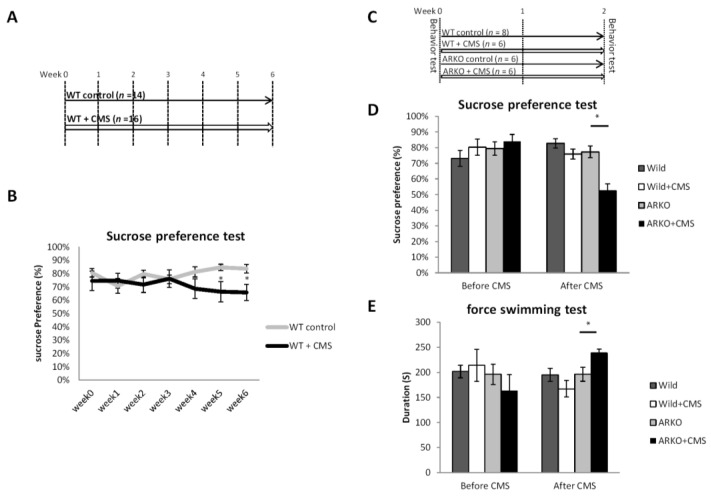
Loss of androgen receptor (AR) accelerated depressive-like behavior in chronic mild stress but does not directly cause depressive-like behavior itself. (**A**) Schematic representation of the mousse experimental setting in (**B**) 2 groups were analyzed, wild type (WT) control (*n* = 14) = C57BL/6 wild type (WT) mice and WT + Chronic mild stress (CMS, *n* = 16) = 6 weeks of CMS exposed WT mice. Effects of CMS on sucrose preference (mean ± SEM of sucrose preference (sucrose/sucrose + water). (**C**) Schematic representation of the experimental setting for (**D**,**E**). Four groups were analyzed, WT control, ARKO: Androgen receptor knock-out (ARKO) mice control, WT + CMS = 2 weeks of CMS on WT mice, ARKO + CMS: 2 weeks of CMS on ARKO mice. (**D**) The sucrose preference test was performed before and after CMS. Mean ± SEM of sucrose preference [sucrose/(sucrose + water)] of 6–10 mice per group. (**E**) The Forced swimming tests were performed within 2 days after the end of CMS. Mean ± SEM of immobility time of 6–10 mice per group. All data presented as mean ± SEM. For the (C) Student’s t-test, unpaired, two-tailed, * *p* < 0.05. For (**E**,**F**), * *p* < 0.05, Student’s t-test.

**Figure 2 cells-08-01021-f002:**
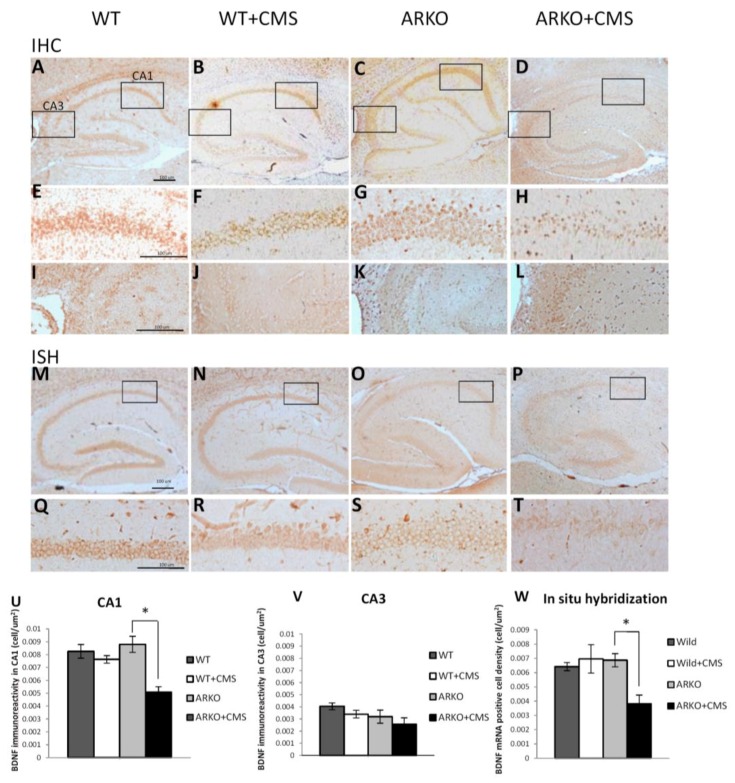
Loss of AR affects BDNF level in response to CMS. (**A**–**L**,**U**,**V**) Immunohistochemistry (IHC) revealed a significant decrease in BDNF protein expression in CA1 area of hippocampus in ARKO mice under CMS for 2 weeks. Mean ± SEM of BDNF protein positive cell density in CA1 area of hippocampus of 6–8 mice per group (**A**–**D**). (**E**–**H**,**U**) BDNF protein expression is down regulated in CA1 area (**E**–**H**). (**I**–**L**,**V**) There is no significant difference in BDNF protein expression in the CA3 area of hippocampus through IHC (**I**,**J**). The (**U**), and (**V**), are quantitative data (mean ± SEM) of (**E**–**H**), (**I**,**J**), respectively. (**M**–**T**,**W**) In situ hybridization (ISH) revealed a significant decrease in BDNF mRNA expression in CA1 area of hippocampus in ARKO + CMS (**M**–**T**). The (**W**) is the quantitative data (mean ± SEM) of (**M**–**T**). * *p* < 0.05, Student’s t-test. Scale bar 100 µm.

**Figure 3 cells-08-01021-f003:**
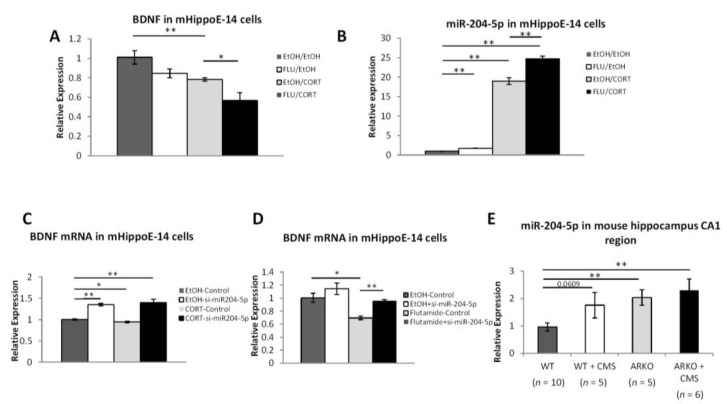
Loss of AR and stress boosts miR-204-5p to inhibit BDNF expression. (**A**) The combination of corticosterone and flutamide greatly down-regulated BDNF mRNA levels in mHippoE-14 cells. (**B**) Corticosterone and flutamide up-regulated miR-204-5p levels in mHippoE-14 cells. (**C**) Expression of miR-204-5p is significantly increased in ARKO mice after receiving 2 weeks of CMS. (**D**,**E**) Knocking down miR-204-5p in mRNA mHippoE-14 cells reverses the effect of corticosterone (**D**) and flutamide (**E**) on down-regulating BDNF mRNA. Data are representative of three independent experiments and are shown as the mean ± SEM compared to controls. * *p* < 0.05 and ** *p* < 0.01.

**Figure 4 cells-08-01021-f004:**
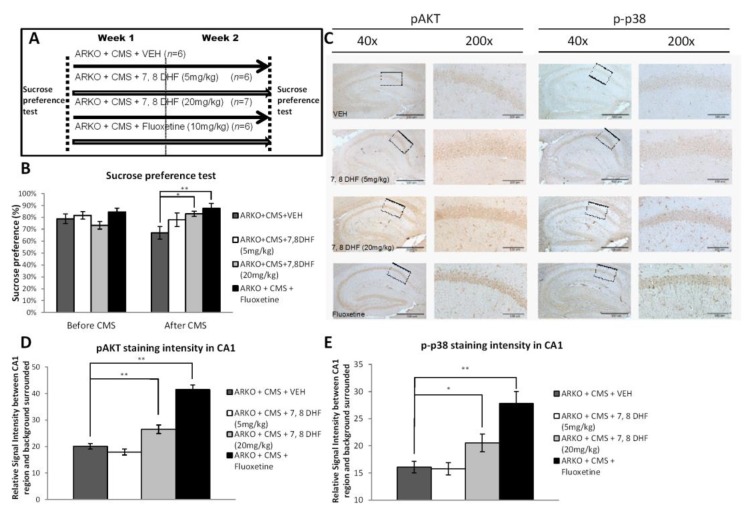
High dose tropomyosin receptor kinase B (TrkB) agonist has effects similar to antidepressants in preventing depressive-like behavior in mice receiving CMS. (**A**) Schematic representation of the experimental setting. Four groups were analyzed, ARKO + CMS + vehicle (VEH) = 2 weeks of CMS on ARKO mice control, ARKO + CMS + 7,8 DHF (5 mg/kg) = 2 weeks of CMS on ARKO mice treated with 5 mg/kg 7,8-dihydroxyflavone, ARKO + CMS + 7,8 DHF (20mg/kg) = 2 weeks of CMS on ARKO mice treated with 20 mg/kg 7,8-dihydroxyflavone, and ARKO + CMS + Fluoxetine (10 mg/kg): = 2 weeks of CMS on ARKO mice treated with 10 mg/kg fluoxetine. (**B**) The Sucrose preference test was performed before and after exposure to CMS. Mean ± SEM of Sucrose preference [sucrose/sucrose + water] of 7–14 mice per group. (**C**–**E**). Immunohistochemistry (C) revealed pAkt staining (D, quantification from c left panels) and p-p38 staining (E, quantification from C right panels) significantly increased in CA1 area after receiving treatment with fluoxetine and 7,8-DHF (20 mg/kg). For (B, D and E), quantitation are mean ± SEM, * *p* < 0.05; ** *p* < 0.01, Student’s t-test. In (c) Scale bar (40×) 500 µm. Scale bar (200×) 100 µm.

**Figure 5 cells-08-01021-f005:**
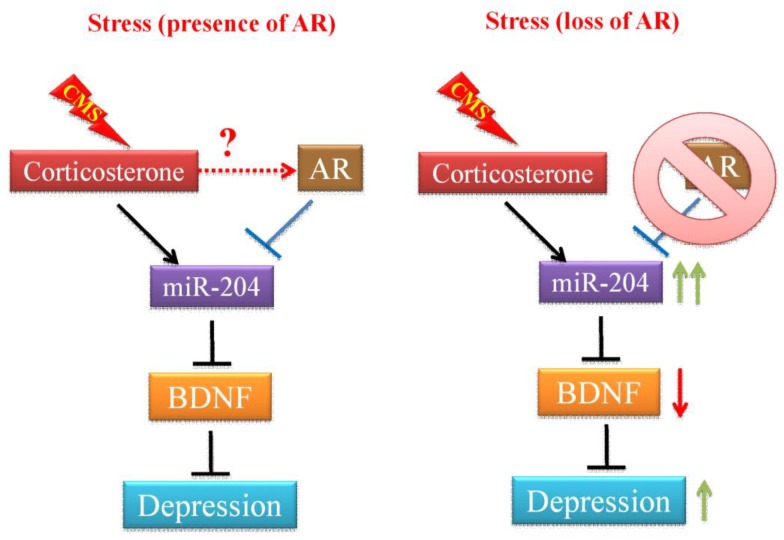
The effects of stress and AR on depression development. In the absence of AR, chronic mild stress (CMS)-mediated corticosterone actions promote the miR-204 expression in the hippocampus cells. The increased miR-204 expression and activity results in a decreased level of brain-derived neurotrophic factor (BDNF), which is a critical regulator of the formation and plasticity of neuronal networks and is abundant in the brain. Decreased BDNF expression has been shown to associate with the status and severity of major depression. In the presence of AR, AR counteracts the expression of miR-204 induced by CMS, therefore, BDNF level is not affected and mice are also more resilient to exhibit depressive-like behaviors under CMS.

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
