# Peer review of "Deficiency in Androgen Receptor Aggravates the Depressive-Like Behaviors in Chronic Mild Stress Model of Depression"

_cells, 2019, doi:10.3390/cells8091021_

Round 1

Reviewer 1 Report

The aim of the present study was to evaluate the mechanism how the decreased androgen receptor influence the development of major depressive disorder (MDD). The submitted research is very interesting, but attention to the following points and questions could potentially improve the manuscript.

1) Based on the results of Figure 1A-1B, the authors planned the schedule of the experiment of Figure 1C-1E. The authors should explain the reason why they selected the experimental period (2 weeks) for Figure 1C-1E in Results (3.1.).

2) It is hard to see most figures, because they are too small.

3) The supplementary Figure 1 (data for neuro2a) is not necessary. The authors should mention their findings as “(data not shown)” in the text (line.327-328).

4) The authors mentioned in detail about the expression of AR, BDNF and PSD-95 in Discussion (line.434-457). In addition to it, the authors should briefly explain the reason why they selected the region (CA1 and CA3 region of hippocampus) to analyze the BDNF expression in Results (3.2.).

5) There is unforced error, such as,

“androgens plus stress” → “loss of androgen receptors plus stress” (line.19 of Introduction)

“2 x 105 cells/mL” → “2 x 105 cells/mL” (line.178 of Introduction)

Author Response

Response to the reviewers’ comments

        Here are our responses to improve the manuscript, in light of the reviewer’s comments. (Point-by-point).

Reviewer 1

The aim of the present study was to evaluate the mechanism how the decreased androgen receptor influence the development of major depressive disorder (MDD). The submitted research is very interesting, but attention to the following points and questions could potentially improve the manuscript.

1) Based on the results of Figure 1A-1B, the authors planned the schedule of the experiment of Figure 1C-1E. The authors should explain the reason why they selected the experimental period (2 weeks) for Figure 1C-1E in Results (3.1.).

Thanks for your suggestion. We have added the references and explained the reason why we schedule the experiment in line 227-228 by mentioning: ”It is well documented that CMS-exposed-mice with C57BL6 background require 3-weeks or longer CMS procedure to develop depressive–like behavior [36]”.

2) It is hard to see most figures, because they are too small.

Thanks for your comments. We have enlarged the size of the figures to make them easier to read.

3) The supplementary Figure 1 (data for neuro2a) is not necessary. The authors should mention their findings as “(data not shown)” in the text (line.327-328).

 Thanks for your suggestion. We have removed the supplementary Figure 1 and mentioned the finding in line 306-309: The results showed that BDNF mRNA expression was decreased under treatment of corticosterone and flutamide in mouse hippocampus cell line mHippoE-14 (Figure 3A) as well as neuroblastoma cell line Neuro-2A cells (data not shown) as reviewer suggested.

4) The authors mentioned in detail about the expression of AR, BDNF and PSD-95 in Discussion (line.434-457). In addition to it, the authors should briefly explain the reason why they selected the region (CA1 and CA3 region of hippocampus) to analyze the BDNF expression in Results (3.2.).

Thanks for your suggestion. We have added a paragraph to explain the reason why we analyzed the expression of BDNF in CA1 in line 256-262 by mentioning: “The expression of AR has been reported to localize mainly in pyramidal neurons from hippocampal CA1 area and scarcely in some neurons from CA3 area and dentate gyrus (35). Moreover, BDNF has also been demonstrated to be significantly decreased in the hippocampus CA1 area of CMS-treated mice [36]. To further dissect whether loss of AR affect the BDNF expression in CMS-treated mice, we examined the BDNF protein expression in the CA1 and CA3 area of hippocampus from WT and ARKO mice under CMS procedures for 2 weeks by using IHC staining (Figure 2A-L).”

5) There is unforced error, such as,

“androgens plus stress” → “loss of androgen receptors plus stress” (line.19 of Introduction)

 Thanks for your correction. We have revised the sentence in line 19 into “While androgen receptor and stress may influence the development of the major depressive disorder (MDD)”.

“2 x 105 cells/mL” → “2 x 105 cells/mL” (line.178 of Introduction)

Thanks for your correction. The typing error in line 169 has been corrected.

Reviewer 2 Report

The manuscript entitled ″Deficiency in androgen receptor aggravates the depressive-like behaviors in chronic mild stress model of depression″ is focusing interesting field of the role of androgen receptor in depressive-like behavior induced by stress and evaluating the mechanisms of AR actions on depression. It should be noticed that the basic idea is worth of careful analysis. However, there are several serious concerns about the manuscript.

General remarks

The combination of evidently different studies (patients, mice, cell culture) is presented in a little bit confusing way. Beside a lot of results presented in the manuscript, I suggest to exclude certain clinical parts (except in Introduction and Discussion) or to reorganize the whole manuscript for better understanding.

Is it necessary to include clinical part in experimental design due to fact that it is well known that AR mRNA expression level is lower in MDD patients? The selection criteria for human subjects involved in the study should be precisely defined as well as exclusion criteria. Why the authors used HAMD-17 scale on MDD patients only? 

Materials and Methods

Parameters obtained in FST should be mentioned in Materials and Methods section.
·Is two weeks sufficient time interval between behavioral testing? It is very questionable, so please include some similar example from literature to support this.
·         The authors should explain the procedures used for genotype determination of WT and ARKO mice.
·         For histology sections, the authors should precisely define the position of sections relative to bregma or other anatomical point.
·         Line 164: the authors should further explain the method used for calculation of positive staining density, possibly with the diagram showing the examined hippocampal regions.
·         The authors examined DAB chromogen signal intensity in IHC experiments.
·         Considering that antigen-antibody reactions are not stoichiometric, so "darkness of stain" does not mean "amount of products", and that DAB chromogen does not follow the Beer-Lambert law (10.1369/jhc.2007.950170), this approach should be minutely discussed. 
·         Line 230: please explain “positive cells were counted by experienced staff“. 

Results

The results of open field test are missing.
·The quality of figures must be improved, and graphs should be uniformed.
·         The authors might consider replacing representative photomicrographs in Figure 2 (C, J, O, P).
·         The figure legends are confusing and should be rephrased.
·         Also, please explain the representative photomicrographs in Supplementary figure 2 (with primary antibody omitted).

Discussion

Age of patients and animals are not matching, it is not comparable. It would be better that experiments were conducted in older animals since the manuscript is focused on age-dependent pathophysiological mechanisms that are not supposed to be found in younger (adolescent, 10 w) animals. Therefore, I suggest that you should at least mention and discuss a very few literature data concerning older animals and the relative expression of BDNF, as well as hippocampal BDNF content in more comparable (to clinical trials results) middle-age (20 w) mice (Brain, Behavior and Immunity, 2019), or older.

Check for the more accurate references, especially concerning the evaluation of hippocampal sex hormones receptors in modulation of depressive-like behavior in animal experimental model (Front Behav Neurosci, 2019).

Pay attention to spelling and grammar through the whole text.

This is high quality manuscript and, after suggested corrections, it would be worth for publication in such an eminent journal.

Author Response

Response to the reviewers’ comments

        Here are our responses to improve the manuscript, in light of the reviewer’s comments. (Point-by-point).

Reviewer 2:

The combination of evidently different studies (patients, mice, cell culture) is presented in a little bit confusing way. Beside a lot of results presented in the manuscript, I suggest to exclude certain clinical parts (except in Introduction and Discussion) or to reorganize the whole manuscript for better understanding.

Is it necessary to include clinical part in experimental design due to fact that it is well known that AR mRNA expression level is lower in MDD patients? The selection criteria for human subjects involved in the study should be precisely defined as well as exclusion criteria. Why the authors used HAMD-17 scale on MDD patients only? 

Thanks for your suggestion. We have removed the clinical parts in the Results and experimental design from the section of Materials and Methods (except in Introduction and Discussion) and reorganized the manuscript for better understanding as reviewer’s suggestion.

Materials and Methods

Parameters obtained in FST should be mentioned in Materials and Methods section.

Thanks for your suggestion. The parameters for FST were added in line124-129:

Forced swimming test.

Glass beakers (13 cm diameter × 24 cm high) were filled with fresh water (23–25 ◦C) to a depth of 12 cm. Mice were placed into the test beaker and were unable to escape or rest by touching the bottom of the beaker. Sessions were video recorded for 5 min and analyzed offline with ethovision XT (Noldus). Mice were scored by visual inspection for immobility defined as motionless floating in the water.

Is two weeks sufficient time interval between behavioral testing? It is very questionable, so please include some similar example from literature to support this.

Thanks for reviewer’s suggestion. According to previous literature (J Neurosci Methods 2011, 195, 200-205, doi:10.1016/j.jneumeth.2010.12.015.), the authors used the repeated forced swim test to reduce the number of animals in evaluating effects of antidepressant. In that work, the interval for FST was 1 week apart (Day 2: test, Day7: retest 1, Day 14: retest 2) and factorial analysis revealed that the test, the retest 1 as well as the retest 2 had variables suitable to detection of antidepressant-like effects of ADT. Therefore, two weeks should be a sufficient time interval between behavioral testing.

The authors should explain the procedures used for genotype determination of WT and ARKO mice.

Thanks for your suggestion. We have added the sentence in the Materials and Methods at line 95-97 by mentioning: “The strains of the mosaic founder mice were C57BL/6 and 129Sv background and Floxed AR/AR female mice were established and the procedures used for genotype determination as described in our previous publication [29,30].“

For histology sections, the authors should precisely define the position of sections relative to bregma or other anatomical point.

Thanks for your suggestion. We have added the sentence to define the position of sections relative to bregma of brain in line 138-139 by mentioning: “BDNF, pAKT and p-p38 staining were performed on 5 μm paraffin brain sections (sagittal, lateral 1.10-1.95mm)”.
·         Line 164: the authors should further explain the method used for calculation of positive staining density, possibly with the diagram showing the examined hippocampal regions.

Thanks for your suggestion. We have added the sentence at line 211-214: “The cell density of BDNF protein immunoreactive cells and BDNF RNA positive cells were counted by using 3-point scale (34). The cell density was defined as positive cell / cells distribution region. The immunoreactive score gives a range as a product of multiplication between positive cells proportion score and staining intensity score (0–3).”

The authors examined DAB chromogen signal intensity in IHC experiments. Considering that antigen-antibody reactions are not stoichiometric, so "darkness of stain" does not mean "amount of products", and that DAB chromogen does not follow the Beer-Lambert law (10.1369/jhc.2007.950170), this approach should be minutely discussed. 

Thanks for your suggestion. We have added paragraph to discuss about the limitation in line 421-424: “There are some limitations in our current study. First, DAB used for immunohistochemistry and in situ hybridization does not follow the Beer-Lambert law that describes the linear relationship between the concentration of a compound and opticaldensity [60]” as reviewer suggested.

Line 230: please explain “positive cells were counted by experienced staff“. 

Thanks for your suggestion. We have revised a paragraph in the section of Materials and Methods in line 212-214 by mentioning: “The cell density of BDNF protein immunoreactive cells and BDNF RNA positive cells were counted by using 3-point scale (34). The immunoreactive score gives a range as a product of multiplication between positive cells proportion score and staining intensity score (0–3).”

Results

The results of open field test are missing.

Thanks for your suggestion. The results of Open field test showed no difference in mice after receiving chronic mild stress as shown in the following figure.

The quality of figures must be improved, and graphs should be uniformed.
·         The authors might consider replacing representative photomicrographs in Figure 2 (C, J, O, P).

Thanks for your suggestion. The quality of figures were uniformed and improved as reviewer suggested.

The figure legends are confusing and should be rephrased.

Thanks for your suggestion. The figure legends were rephrased as reviewer’s suggestion.

Also, please explain the representative photomicrographs in Supplementary figure 2 (with primary antibody omitted).

Thanks for your suggestion. In our previous submission, we are asked by reviewers to add the data about photomicrographs in Supplementary figure 2 with primary antibody omitted to show the extent of unspecific/background signal.

.

Discussion

Age of patients and animals are not matching, it is not comparable. It would be better that experiments were conducted in older animals since the manuscript is focused on age-dependent pathophysiological mechanisms that are not supposed to be found in younger (adolescent, 10 w) animals. Therefore, I suggest that you should at least mention and discuss a very few literature data concerning older animals and the relative expression of BDNF, as well as hippocampal BDNF content in more comparable (to clinical trials results) middle-age (20 w) mice (Brain, Behavior and Immunity, 2019), or older.

Thanks your suggestion. We fully agree with the reviewer that the age of patients and animals are not matching in our study and have discussed the limitation of our data to show that AR is related to depression in young men and male adolescent in line 421-431: “Second, age of mice in this study does not match human data. While major depressive disorder in young men [9] and male adolescent [61] were reported to be important and associated with AR, our study is focused on age-dependent pathophysiological mechanisms. The aging process can lead to the impairment of brain functions and a decrease in whole brain volume in elderly when compared with young adults, especially in brain regions related to cognition has been reported (62), it would be better that our experiments could be reproducibly conducted chronic mild stress protocol in older animals with relative low level of testosterone and BDNF expression.”

Check for the more accurate references, especially concerning the evaluation of hippocampal sex hormones receptors in modulation of depressive-like behavior in animal experimental model (Front Behav Neurosci, 2019).

Thanks for your suggestion. We have added the reference to update the evaluation of hippocampal sex hormones receptors in modulation of depressive-like behavior in animal experimental model in line 58-59 by mentioning: “Furthermore, AR/ERα expression index in the hippocampus were reported to regulate depressive-like behavior in rats [15]” as reviewer’s suggestion.

Round 2

Reviewer 2 Report

I suggest that the authors should provide more accurate information for an interconnection between BDNF levels and behavioral testing outcome, in Discussion section, such as: More specific, it has been recently reported that hippocapal BDNF relative gene expression in mice positvely correlated with specific neuron's populations in hippocamous, such as GABA-containing (Stajic et al., 2019), that are involved in depressive level regulation (Oh et al., 2019).

I suggest that manuscript should be accepted after these corrections, and there is no need for another review.

Author Response

Reviewer comments:

I suggest that the authors should provide more accurate information for an interconnection between BDNF levels and behavioral testing outcome, in Discussion section, such as: More specific, it has been recently reported that hippocapal BDNF relative gene expression in mice positvely correlated with specific neuron's populations in hippocamous, such as GABA-containing (Stajic et al., 2019), that are involved in depressive level regulation (Oh et al., 2019).

Answer: Thank you for your comments ans suggestion. We have added the information for an interconnection between BDNF levels and behavioral testing outcome in Discussion section by mentioning " In agreementwiththe previous findings that the BDNF protein level in the CA1of the hippocampus is decreased under CMS [59]and the relative gene expression of hippocampal BDNF in mice is positively correlated with specific neuron's populations in hippocampus, such as GABA-containing neurons [60]which are involved in depressive level regulation [61]" in line 419-423 as reviewer suggested.

This manuscript is a resubmission of an earlier submission. The following is a list of the peer review reports and author responses from that submission.

Round 1

Reviewer 1 Report

Making use of the model of androgen receptor knock-out mouse (ARKO) developed previously in the Authors’ lab and taking into account multiple clues in the literature linking stress/depressive behavior with androgen/TrkB/BDNF level, including data showing that BDNF is a target of miR-204, Hung et al. investigated depressive behavior in ARKO mice. They found that ARKO mice are more susceptible to depressive behavior and, after stress, exhibit lower BDNF level in CA1, but not CA3, of the hippocampus. ARKO mice also have higher level of miR-204 in CA1. Activation of TrkB receptors in ARKO mice counteracts the effect of stress. Using mHippoE-14 cells the Authors showed that the decrease in BDNF following treatment with corticosterone and an antiandrogen drug, flutamine, can be reversed by miR-204-5p siRNA. They conclude that the lack of androgen receptor results in higher miR-204 level in CA1 which, in turn, reduces  BDNF and activation of AKT and MAPK signaling. A study measuring AR mRNA in male patients with MMD is also included.

Taking into account how much has already been known on the subject the findings are not very original; nonetheless, this research directly documents that knock-out of the androgen receptor results in lower BDNF and increased miR-204 level in CA1 of the hippocampus and in higher propensity to depressive behavior in ARKO mice following CMS.

1. I was very astonished to find that the Authors examined pMAPK staining (Fig. 4) using anti-pMAPK antibody!! The term “MAPK” (mitogen –activated protein kinase(s)) refers to a family of kinases that includes ERK, JNK and p38 kinases, so the term “anti-pMAPK antibody” makes no sense. When verified on the manufacturer’s page, the “ anti-pMAPK antibody (10 μg/mL Bioss #bs-2210R”(according to Authors), appears to be a “p38 MAPK (Thr180 + Tyr182) Antibody. catalog : bs-2210R”, which clearly recognizes the phosphorylated form of p38 kinase. Weren’t the Authors aware which MAPK they study? This is a gross factual error which must be corrected. The term MAPK may, optionally, be preserved in the title, meaning MAPK in the broad sense (see above), even if only p38 was examined. However,  the term “..pAKT/pMAPK signaling” should be substituted by “..AKT/MAPK signaling” .

2. Many of the results presented in this work rely on IHC and HIS staining quantified by a computer program. In my opinion control reactions i.e. without primary Ab should be included for each studied antigen to show the extent of unspecific/background signal.

Minor points:

1. Fig 3. The description of parts C,D,E is mixed up. Furthermore the Authors state: “Expression of miR-204-5p is significantly increased in ARKO mice after receiving 2 weeks of CMS.” when there is no statistically significant difference between stressed and control ARKO mice. Also, it would be interesting to discuss why, even though non-stressed ARKO mice have higher miR-204-5p level in CA1, the BDNF level is the same as in controls and drops down only after CMS?

2. Figure lettering and size should be increased to improve their readability; the bar color code in Fig.1 and 3 should be changed to resemble that in Figs 2 and 4.

3. Fig. 5 mentioned in Discussion is nowhere to be found?

4. English language should be polished to avoid phrases such as “could leads”,” BDNF mRNA treated with corticosterone in mHippoE-14 cells”, “mice treated with or without CMS”.  Several sentences e.g. “Mouse exposure to stress has been linked to the decreased brain-derived neurotrophic factor (BDNF) expression in the serum of patients with major depression[15, 16] and in the hippocampus from suicide victims[17-19]” should be rewritten to be become comprehensible.

Reviewer 2 Report

In this paper, the authors aimed to investigate how  AR might function via altering the expression of miR-204-5p to modulate the brain-derived neurotrophic factor (BDNF) expression to influence depressive-like behaviors in the mice under the CMS. Although this is an interesting study, the reviewer has some major concerns detailed below.

Major comments

The mouse strain for WT animals and the background strain for the ARKO mice is not indicated in the manuscript and must be added.

Page 2, second paragraph: This sentence does not make much sense and should be rewritten. “Mouse exposure to stress has been linked to the decreased brain-derived neurotrophic factor (BDNF) expression in the serum of patients with major depression[15, 16] and in the hippocampus from suicide victims[17-19].”

The authors selected the mHippoE-14 cell line to perform their siRNA in vitro studies and validate the fact that those miRNA are involved in male rodents (and by extension, humans, in MDD phenotype). However, mHippoE-14 cell line originates from embryonic female hippocampus, as determined by the lack of expression of the male chomosome marker, SSTY1 (Neuroscience. 2010 Sep 29;170(1):54-66. doi: 10.1016/j.neuroscience.2010.06.076. Epub 2010 Jul 7. Estrogen receptor α and G-protein coupled receptor 30 mediate the neuroprotective effects of 17β-estradiol in novel murine hippocampal cell models. Gingerich S1, Kim GL, Chalmers JA, Koletar MM, Wang X, Wang Y, Belsham DD.) This is a serious concern as far as interpretation of the data and extrapolations on the pathways recruited. These experiments must be repeated in male cell lines to confirm the data before publication.

Other works have shown that an increase in miRNA 204 in hippocampus neuronal cells is associated with senescence during aging. Androgen deficiency or deprivation has also been shown to induce senescence in cancer cells, but other neuronal types as well. Could it be that the absence of androgens on the hippocampus both in vivo and in vitro and the subsequent upregulation of miRNA 204 is essentially related to neuronal cells senescence? As shown in Fig 3E, miR-204-5p is significantly increased in naïve ARKO mice and is slightly increased after receiving 2 weeks of CMS, without reaching statistical analysis. Similarly, in Fig 3B, miRNA 204 levels are significantly increased following flutamide treatment. Because the only availability for androgens in cell culture would be those issued from FBS (shown to be low and similar to castrated males), the changes in miRNA 204 expression in vitro would not be as dramatic as the ones seen in vivo (especially compared to the huge increase provoked by corticosterone treatment).

Also, the correlation suggested by the authors between varying BDNF levels (mRNA or protein) and expression of miR-201-5p in hippocampus CA1 region in WT and ARKO mice exposed or not to CMS do not match when one looks closely at Fig 2U, 2W and 3E. The relation at play do not appear to be that simplistic and it is hard to be convinced about the data.

Justification for the dose of corticosterone? How is this clinically relevant and comparable to CMS model?

Legend Fig 1: the group numbers are not in line with the ones indicated in the figures. This should be corrected.

Fig 1C: the line for control mice is not visible. The figure must be redone.

Fig 1E and 1F: the colors of the columns must be changed and unique for each group of mice so that there is no possible confusion between groups.

Fig 3C and 3D: the titles of the graphs do not make much sense. Do the authors mean BDNF mRNA levels in mHippoE-14 cells treated with corticosterone/flutamide?

Minor comments

Page 5, result paragraph 3.1, line 1: Misspelling “investigate”

Page 5, result paragraph 3.1, line 8: Misspelling “procedure”

Page 9, last line: misspelling chornic variant